# Influences of Substrate Grain Size on the Burrowing Behavior of Juvenile *Meretrix meretrix*

**DOI:** 10.3390/ani12162094

**Published:** 2022-08-16

**Authors:** Changsheng Zhang, Suyan Xue, Jiaqi Li, Jinghui Fang, Lulei Liu, Zhanfei Ma, Wenhan Yu, Haonan Zhuang, Yuze Mao

**Affiliations:** 1Yellow Sea Fisheries Research Institute, Chinese Academy of Fishery Sciences, Qingdao 266071, China; 2Chinese Academy of Agricultural Sciences, Beijing 100081, China; 3Marine Ecology and Environmental Science Laboratory, Pilot National Laboratory for Marine Science and Technology, Qingdao 266237, China

**Keywords:** *Meretrix meretrix*, substrate grain size, substrate preference, behavioral characteristics, burrowing ability

## Abstract

**Simple Summary:**

*Meretrix meretrix* lives in substrates at a depth of 1–20 cm; the substrate has an influence on its growth, survival, living habits, and behavioral characteristics. In this study, we investigated the effects of different grain size substrates on substrate preference, burrowing ability, and behavior during the substrate selection process of *M. meretrix*. These results indicated that juvenile *M. meretrix* had a significant preference for and a stronger burrowing ability in fine sand. As the substrate grain size increased, the burrowing ability and preference of *M. meretrix* decreased, and these bivalves showed behavioral characteristics such as a prolonged selection time and an increased percentage of movement. In addition, by observing the substrate selection behavior of *M. meretrix*, we divided the selection process of the substrate by juvenile *M. meretrix* into four stages: preparation, selection, burrowing, and end stages.

**Abstract:**

The substrate is the key environmental factor that affects the growth, survival, population and distribution of dwelling mollusks in mudflat settings. To clarify the effect of the substrate grain size on soft substrate preference, burrowing ability and behavior during the selection process of juvenile *Meretrix meretrix*, four different grain size substrates (coarse sand, medium sand, fine sand, and natural substrate) were set up for comparison. The results indicated that: (1) the burrowing ability of juvenile specimens in fine sand was the strongest; (2) the degree (from high to low) of the juvenile’s preference for the four substrates was in the order of fine sand > natural substrate > medium sand > coarse sand; and (3) the selection process of the substrate by the juveniles could be divided into four stages: preparation, selection, burrowing and end stages. These stages showed the behavioral characteristics of a longer selection time and higher percentage of movement in coarse sand. Therefore, our results demonstrated that sea areas or ponds with fine sand as the main component are more suitable for stock enhancement with *M. meretrix*. These results provide basic data for habitat selection and suitability evaluations for the aquaculture of *M. meretrix*.

## 1. Introduction

Habitat selection is a complex process, influenced by a combination of biotic factors (morphology, individual health, behavior) and abiotic factors (aquatic environment and habitat structure) at different temporal and spatial scales [1,2]. Marine organisms decide when and where to settle based on their endogenous state and specific physical or chemical cues in the environment to maximize adaptation [3,4]. For the zoobenthos, the substrate is one of the decisive factors in their habitat selection process, and the substrate environment significantly affects their growth, survival, distribution, and habitat [5,6]. It has been shown that the grain size and shape of the soft substrate can cause differences in the overall substrate shear strength, cohesion, water content and sediment microtopography, which in turn affect the suitability of the substrate as a habitat [7,8,9]. For example, *Venerupis philippinarum* had different survival and growth rates in four different grain sized substrates, and medium sand with grain sizes ranging from 180 to 335 μm was more suitable for their growth and survival [6]. Compared to mud substrates, sand substrates have larger particles which to some extent hinders the search for predators and can provide safer shelter for *Mercenaria mercenaria* [10]. The size of the substrate grain size affects the burrowing ability and metabolic rate of *Donax trunculus*, and they have a faster burrowing rate and higher metabolic rate in small grain size substrates when compared to large grain size substrates [11].

The substrate largely determines the survival conditions of the zoobenthos, and the ability to identify and select a suitable substrate is therefore essential [12]. Current research shows that some benthic shellfish can recognize different types of substrates and choose accordingly by weighing the benefits (food availability and survival conditions) and risks (risk of predation and competitive pressure) [13]. For example, *Scapharca broughtonii* [14], unionid mussels [15], and *freshwater mussels* [16] possess the ability to actively select their habitat and show different preferences for different substrates. The behavioral characteristics (behavior, activity rhythm, locomotion) of benthic shellfish are formed by long-term evolutionary processes as a result of the interaction between organisms and their environment, and therefore the selection of the substrate is the expression of an adaptation to that substrate type [11]. In addition, behavioral characteristics such as burrowing speed and horizontal or vertical movement ability can reflect the degree of adaptation to the substrate [14,17]. It is intuitive and rapid to investigate the adaptation of marine organisms to the substrate through behavioral responses, and this method has been widely used for species such as fish [13,18], crabs [19,20], and shrimps [2]. However, studies on the behavior of benthic shellfish are not yet thorough, and such studies can be used to screen for suitable substrate types for benthic shellfish and provide data for behavioral studies on shellfish.

*Meretrix meretrix* (Mollusca, Bivalvia) thrives in a wide temperature and salinity range, and lives in soft substrate at a depth of 1–20 cm, the shells have a triangular ovoid appearance and are mostly yellow and brown (Appendix A) [21]. *M. meretrix* is widely distributed along the coast of China, with abundant resources in the Liaoning, Shandong, Jiangsu, and Fujian Provinces [22,23]. It is one of the main commercial bivalves in China, and the annual production is approximately 3.5 × 10^5^–4.0 × 10^5^ t, accounting for more than 90% of the world production [24]. For example, the average annual production of *M. meretrix* in a town in Zhejiang Province is about 500 kg/667 m^2^, with a total annual production of 3200 t and an export price of up to $2500 per t, which has good economic benefits [25]. However, due to overfishing, and habitat overexploitation, the natural recovery of *M. meretrix* in China has been severely damaged [26]. Thus, stock enhancement, through the release of hatchery-reared juveniles, has become one of the main methods to restore the *M. meretrix* resources [27,28]. Selecting high-quality species and ensuring the physiological health of the juveniles are the prerequisites for clam enrichment release [29,30]. Additionally, selecting suitable release sites according to the substrate adaptability of juvenile *M. meretrix* has practical importance to improve the quality of the reared individuals for stock enhancement. Previous studies have shown that substrate grain size is considered to be one of the key factors that affect the survival and distribution of *M. meretrix*, and in the field, *M. meretrix* has been shown to escape harsh substrate environments by migrating or possessing the ability to identify and select suitable substrates [21,26,31]. However, the substrate preference of juvenile *M. meretrix* has not been studied under laboratory conditions, and there are few studies concerning how the substrate size affects the behavioral characteristics of juvenile *M. meretrix*. In this study, we used experimental ecology to investigate the adaptability of juvenile *M. meretrix* to substrates by observing their selectivity for four different grain sizes of substrates and their behavioral characteristics during the selection process to clarify the suitable substrate types for juvenile *M. meretrix*. The artificial culture process of *M. meretrix* is divided into parental maturation promotion and spawning, larvae cultivation, intermediate cultivation of juvenile *M. meretrix*, mudflat or pond stocking, and culture [32]. Thus, clarifying the substrate adaptability of *M. meretrix* can provide a reference for habitat selection during intermediate pond cultivation and stock enhancement in the mudflats.

## 2. Materials and Methods

### 2.1. Acquisition of Experimental Animals and Substrates

The ethical regulations concerning the use of experimental animals were followed (see the Statement of the Ethics Committee in the Appendix A). The experimental juvenile *M. meretrix* were taken from the Hekou District, Dongying City, Shandong Province, in October 2021 and transported to the indoor laboratory at Langya Base, Yellow Sea Fisheries Research Institute. Then, healthy and vigorous individuals were selected with a wet weight of 0.36 ± 0.1 g, a shell length of 10.8 ± 1.4 mm, and without shell damage and were temporarily reared in a recirculating water culture system for seven days. The water temperature was 20 ± 1 °C, seawater salinity was 30 ± 1‰, pH was 8.0 ± 0.2, and dissolved oxygen was 7.0 ± 0.5 mg/L, and there was daily water exchange of about a third. Also, there was morning and evening feeding with bait (diatoms and *Nannochloropsis oceanica*, and after feeding the water column algae concentration was 2 × 10^4^ cells/mL).

The substrate that was used in the experiment was taken from the natural habitat of the clam in Laizhou Bay, Weifang City, and was soaked in potassium permanganate for 24 h for sterilization. Then, it was dried and sieved into coarse sand (grain size of 500–2000 μm), medium sand (grain size of 250–500 μm), and fine sand (grain size of 63–250 μm) based on the Wentworth scale [33]. Four substrates with different grain sizes were used in this study (Appendix A provides a picture of the four substrates), i.e., coarse sand, medium sand, fine sand, and unscreened natural habitat substrate (hereafter referred to as “natural substrate”). The natural substrate was analyzed using a Mastersizer 3000 laser particle size meter (Malvern Instruments Ltd., Malvern City, UK) and the results are shown in Table 1.

### 2.2. Video Device

The camera unit mainly consisted of a monitor (546.1 mm), a hard disk recorder (eight channels), and a camera (4 MP), which was mounted above the experimental device to ensure clear and complete observation of the experimental animals and device (Appendix A provides a picture of the entire experimental apparatus in the laboratory).

### 2.3. Experiment Ι

The device that was used in this experiment is shown in Figure 1. Four plastic boxes of the same size (coarse sand, medium sand, fine sand, natural substrate) were lined with 8 cm thick substrates of different grain sizes and placed completely in the recirculating water system. *M. meretrix* was placed evenly and lightly on the surface of the substrate (to avoid stirring up the sand and dust, which would affect the observations), and fifty *M. meretrix* were placed in each substrate area. Then, the experiments were conducted for 24 h and repeated five times, during which continuous recordings were taken with a video camera device. The burrowing time (the time taken from the beginning of the clam’s vertical shell to it being completely buried in the substrate) of *M. meretrix* in different substrates was recorded and was used to measure *M. meretrix*’s burrowing ability.

### 2.4. Experiment Ⅱ

In this experiment, the four substrates were combined in pairs. The preference of the juvenile *M. meretrix* for the substrate was evaluated by observing their selection for two substrates in each combination. The device that was used in this experiment is shown in Figure 2. The plastic box (Figure 2-Ι) was lined with two different grain size substrates (Figure 2-Ⅱ) of 8 cm in thickness and an equal area per substrate combination, with the substrates including six combinations: “coarse sand-medium sand”, “coarse sand-fine sand”, “coarse sand-natural substrate”, “medium sand-fine sand”, “medium sand-natural substrate”, and “fine sand-natural substrate”. For the experiment, ten juvenile *M. meretrix* were placed at the substrate junction (Figure 2-Ⅲ), and to avoid selective preference due to different foot directions [14], the *M. meretrix* were placed with five directed to the left and five directed to the right. The *M. meretrix* that moved into a substrate and burrowed into it completely were considered to select this substrate, and individuals that did not move or burrow were considered unselected. The experiments were conducted in a recirculating water system, and the experiment was conducted for 24 h and repeated five times while recording continuously with a video recording device. The percentage of *M. meretrix* selection for each substrate was counted using the video recording, and the percentage of selection (%) = the number of specimens selecting that substrate/the total number of specimens placed × 100%.

### 2.5. Experiment Ⅲ

In this experiment, the four substrates were combined. The preference of the juvenile *M. meretrix* for the substrate was evaluated by observing their selection for the four substrates. The device that was used in this experiment is shown in Figure 3. The plastic box (Figure 3-Ι) was divided into four areas of equal size (Figure 3-Ⅱ), which were filled with four different grain size substrates, and a transparent plastic disk with a diameter of 8 cm was fixed in the middle position (Figure 3-Ⅳ). For the experiment, thirty juvenile specimens were randomly and evenly placed in the plastic trays to observe the selection behavior of *M. meretrix* for the substrate. The positions of the four substrates were changed randomly before each experiment to prevent the preference of the specimens for a certain location in the plastic box from affecting the accuracy of the results. The number of specimens that were selected for each substrate was counted, and the percentage of selection was calculated, with the same observation and calculation method as in experiment Ⅱ.

### 2.6. Experiment Ⅳ

In experiments Ⅱ and Ⅲ, the preference of *M. meretrix* was assessed by determining their selectivity for substrates, and the results showed that the *M. meretrix* showed a different preference for the four substrates. Experiment Ⅳ was designed to further determine the preference of *M. meretrix* for the substrates, and the specimens were placed in a certain substrate, and whether they actively moved to the other substrates and make a selection was observed. The device that was used in this experiment is shown in Figure 4. The plastic boxes were divided into eight areas of equal size, and two substrates were filled in each area in an alternating arrangement as a substrate combination. The two substrates in each combination were called “Substrate A” and “Substrate B”, and there were twelve combinations (Table 2). In the experiment, five specimens were placed evenly and undirectedly in each combination of “Substrate A”, and the selection behaviors of *M. meretrix* for “Substrate A” and “Substrate B” were observed. The experiments were performed in a circulating water system for 24 h and were repeated five times, and they were recorded continuously with a camera device. The percentage of selection (%), selection time (min), and percentage of movement (%) were counted at the end of the experiment. The calculation for the percentage of selection was the same as in experiment Ⅱ. The selection time (min) was the duration between the initial placement time of the specimens and the completion of the initial burrowing. Then, the percentage of movement (%) = the number of movements (the inconsistency between the burrowing area and the initial placement area was considered to be movement)/the total number of clams placed × 100%.

### 2.7. Statistical Analyses

The data were analyzed and plotted using Excel 2016 and R 3.6.3 statistical analysis software (Lucent Technologies, Murray Hill, NJ, USA), and these results were presented as Mean ± standard error. The burrowing time, percentage of selection, selection time, and percentage of movement of *M. meretrix* were analyzed using a one-way analysis of variance (*p* < 0.05). If the differences were significant, Duncan’s multiple comparisons were conducted to test the differences between the groups. Additionally, independent-sample *t*-tests were applied for comparisons between two samples (the percentage of selection when the four substrates were combined in pairs), and *p* < 0.05 was used as the criterion for a significant difference between the different treatment groups.

## 3. Results

### 3.1. Burrowing Ability of Juvenile Meretrix meretrix in Different Grain Size Substrates

In this study, the substrate grain size had a significant effect on the burrowing time of the juvenile specimens (*p* < 0.05). The burrowing time increased with the increase in grain size, and the longest burrowing time in coarse sand was 3.75 ± 0.15 min, which was significantly higher than that of the other three groups (*p* < 0.05). The burrowing time of the medium sand group was 3.10 ± 0.13 min and that of the natural substrate group was 3.05 ± 0.11 min, with no significant difference between them (*p* > 0.05), and both were significantly higher than that of the fine sand group (*p* < 0.05). The burrowing time in the fine sand was 2.46 ± 0.09 min, which was the shortest time (Figure 5).

### 3.2. The Preference of the Juvenile Meretrix meretrix for Different Grain Size Substrates

In experiment Ⅱ, *M. meretrix* showed no significant preference (*p* > 0.05) for the “fine sand-natural substrate” combination when the four substrates were combined in pairs, while they showed a significant preference for the other five combinations (*p* < 0.05). Among the six combinations, the least preferred was coarse sand (40%, 40%, and 32%), followed by medium sand (56%, 36%, and 34%), fine sand (60%, 60%, and 46%), and natural substrate (62%, 58%, and 48%; Figure 6).

In experiment Ⅲ, when the four substrates were selected simultaneously, *M. meretrix* showed the lowest preference for coarse sand (6%), which was significantly lower than that of the other three groups (*p* < 0.05), and was followed by medium sand (13%), fine sand (28%), and natural substrate (25%). There was no significant difference between these three groups (*p* > 0.05; Figure 7).

In experiment Ⅳ, when “substrate A” was coarse and medium sand, the average percentages selected were 62.7 ± 4.5% for coarse sand and 68.0 ± 2.5% for medium sand, and there was no significant difference between the two groups (*p* > 0.05). However, they were significantly lower than that of the fine sand group (89.3 ± 2.7%) and the natural substrate group (88.0 ± 2.5%; *p* < 0.05; Figure 8).

Substrate grain size had a significant effect on the preference of the juvenile *M. meretrix* (*p* < 0.05), and the results of the selection for the different substrate grain sizes in the three experiments were similar. Also, the degree (from high to low) of their preference was in the following order: fine sand > natural substrate > medium sand > coarse sand.

### 3.3. Behavioral Characteristics of Juvenile Meretrix meretrix in the Process of Substrate Selection

#### 3.3.1. The Selection Process of the Substrate for *Meretrix meretrix*

Regarding the results of the study on the burrowing behavior of *M. meretrix* by Zhang et al. [28], combined with the observation of the substrate selection and burrowing behaviors of the juvenile specimens in this experiment, we divided the substrate selection process into four stages, which were the preparation, selection, burrowing, and end stages (Figure 9). The preparation stage lasts from when the clam naturally falls onto the surface of the substrate to when it starts to extend its siphons and foot, while it maintains a closed shell and is stationary. During the selection stage, *M. meretrix* opens its shell slightly and extends its water siphons and foot. If the nearby substrate meets its survival needs, *M. meretrix* anchors in the substrate with its foot, starts the vertical shell action (*M. meretrix* inserts its foot and anterior shell into the substrate, with the posterior shell pointing upward, and drives its body perpendicular to the substrate), and starts burrowing. If the current substrate is not suitable for survival, the *M. meretrix* starts moving in an undirected manner with the vertical shell, looking for a suitable substrate through continuous exploration, and stops moving after finding a suitable substrate and starts burrowing. During the burrowing stage, *M. meretrix* uses its foot to anchor in the substrate and digs with continuous elongation and contraction of its foot, driving its body to swing back and forth and gradually entering the substrate layer until it is completely submerged in the substrate. In the end stage, the clam is completely burrowed into the substrate and is stationary with the shell maintained upright in the substrate.

#### 3.3.2. The Selection Time of *Meretrix meretrix*

In experiment Ⅳ, the substrate grain size had a significant effect on the selection time of *M. meretrix* (*p* < 0.05). When *M. meretrix* selected the four substrates, the selection time for the coarse sand was the longest (21.7 ± 0.28 min), which was significantly higher than that of the other three groups (*p* < 0.05). The selection time for the other three substrates was shorter, namely, the selection times were 14.3 ± 1.01, 15.1 ± 1.40, and 13.9 ± 1.07 min for the medium sand, fine sand, and natural substrate, respectively, with no significant difference between the three groups (*p* > 0.05; Figure 10).

#### 3.3.3. The Percentage of Movement for *Meretrix meretrix*

In the experiment Ⅳ, the substrate particle size had a significant effect (*p* < 0.05) on the percentage of movement for the juvenile *M. meretrix*. When *M. meretrix* was placed in coarse sand, the percentage of individuals showing movement was 53.3 ± 4.2%, and the percentage of movement was significantly higher than that of the other three groups (*p* < 0.05). When the specimens were placed in the other three substrates, the percentage of movement was significantly lower (*p* < 0.05), which was 29.3 ± 5.4% for the medium sand, 25.3 ± 3.9% for the fine sand, and 28.0 ± 7.4% for the natural substrate, and there was no significant difference (*p* > 0.05) between the three groups (Figure 11).

## 4. Discussion

### 4.1. Burrowing Ability

When benthic shellfish are exposed on a beach, they must burrow quickly and efficiently to avoid the hazards of waves and strong turbulence due to prolonged exposure [11]. The shorter the burrowing time of benthic shellfish, the fewer chemical cues remain in the substrate and the lower the probability of being found and captured by predators [34,35]. Additionally, tides and wind waves can destroy the pores that benthic shellfish leave on the surface of the substrate for seawater exchange, feeding, and respiration, and a rapid excavation efficiency allows them to rebuild pores more quickly [36,37]. The most important abiotic factor that affects their digging speed is the substrate nature. The physical properties of the soft substrate, such as particle size, particle shape, and shear strength, affect their burrowing speed, so indicators such as the burrowing speed and burrowing time of the organisms in different substrates are often used to measure the degree of adaptation to the substrate [38,39].

De la Huz et al. [11] found that *D. trunculus* with shell lengths between 25 and 45 mm took longer to be submerged in gravel substrates but could quickly complete sand dives in fine and medium sand substrates. Also, Fiori et al. [40] showed that the digging speed of *Amarilladesma mactroides* was greatly inhibited when the substrate grain size was larger than 1000 μm or smaller than 62 μm, and the fastest sand diving speed was observed when the substrate grain size was between 63 and 500 μm. This study showed that the burrowing time (from short to long) of *M. meretrix* could be listed in the order of fine sand > natural substrate > medium sand > coarse sand, and the burrowing time increased with the increase in the substrate grain size. The substrate grain size will affect bivalve burrowing to a certain extent since the instability of a large grain size substrate will reduce the burrowing ability of the bivalve, and a small grain size substrate with a high density and stability is more suitable for anchoring and burrowing [41].

However, the tightness of the small grain size substrate will cause the shear strength of the substrate to increase, which has a negative impact on the burrowing of shellfish [3,42]. However, the present study found that the burrowing time of *M. meretrix* in fine sand with a higher shear strength was the shortest, which may be because *M. meretrix*’s burrowing was carried out using both morphological and behavioral effects [43]. During burrowing, *M. meretrix* pushes the surrounding sand to both sides through the back-and-forth oscillation of its shell, effectively reducing the tightness of the substrate. Also, the streamlined body and symmetrical, smooth, and delicate shell of *M. meretrix* greatly reduce the friction between the substrate grain and the body [43,44], which counteracts the negative effects of shear strength through the combination of various adaptive mechanisms such as its behavior, morphology, and shell pattern, so that *M. meretrix* has a strong burrowing ability in fine sand. Similar phenomena also exist in other zoobenthos, for example, *S. broughtonii* use strong foot motility to overcome the shear strength of small grain size substrates [14]. Furthermore, *Perinereis aibuhitensis* can mitigate the friction between the organism and the substrate by secreting mucus [39]. This may be a survival strategy of zoobenthos; to complete burrowing faster and improve their chances of survival, different zoobenthos have evolved unique ways of burrowing according to their survival environment.

### 4.2. Substrate Preference

Studies have shown that substrate grain size plays an important role in the habitat selection and distribution of zoobenthos, and some individual organisms can even distinguish and select the substrate based on the grain size [16]. When Huehner et al. [15] observed the preference of four unionid mussels for substrates under natural and laboratory conditions, they found that *Anodonta grandis* showed a high preference for substrates with smaller grain sizes, while *Lampsilis radiata radiata* and *Lampsilis radiata luteola* showed a preference for sandy substrates in both the field and laboratory, and *Elliptio dilatata* showed no significant preferences for substrates. Sun et al. [39] found that *Perinereis aibuhitensis* has an obvious ability to select suitable substrates, preferring the mud and fine sand substrates with smaller grain sizes. In this study, the preference of *M. meretrix* was significantly influenced by the substrate grain size, which was negatively correlated overall with the substrate grain size, with a preference for fine sand and substrates dominated by fine sand. Wang et al. [21] pointed out that the substrate for *M. meretrix* culture should be fine sand and chalky sand. Then, Wang et al. [26] investigated the distribution characteristics of shellfish in the Geligang district of Liaoning Province and found that the distribution of *M. meretrix* had a highly significant negative correlation with the substrate grain size. Additionally, Henmi et al. [31] found that the density of juvenile *M. meretrix* in the mudflats of Shirakawa Prefecture, Japan, was higher in areas with a small grain size substrate. Under natural conditions, juvenile *M. meretrix* are mainly distributed in marine areas with a smaller substrate grain size, which is consistent with our results.

The substrate preference of *M. meretrix* can be considered an adaptation to the substrate, where differences in substrate grain size cause differences in the physical, chemical, and biological factors in the depositional environment, which in turn affect the preference of *M. meretrix*. In terms of the physical properties, compared with coarse sand and medium sand, fine sand substrates have smaller crevices, lower surface tension, stronger cohesion, and a more stable structure [36,39], which reduces the risk of *M. meretrix* being washed out by turbulence and exposed to the beach after burial. *M. meretrix* can stabilize the densely accumulated fine sand with mucus and establish pores that connect to the sediment surface for respiration and feeding, while the pores in coarse sand substrates are harder to maintain [45]. Compared to large-grained substrates, small-grained substrates recover more easily after *M. meretrix* burrowing, smaller crevices reduce the spread of chemical cues, and fine sand provides safer shelter in the presence of predators [10,46]. In terms of chemical factors, the level of organic matter content is largely influenced by the substrate grain size; the organic matter of the larger grain size substrates is lost easily, and for the fine sand substrates with low exposure, organic matter is easier to accumulate, which ensures the energy demand of the zoobenthos [9]. The redox potential in the substrate is also correlated with the substrate grain size since smaller grain size substrates are less influenced by the overlying water and have a higher redox potential, this makes it easier for reducing bacteria to metabolize, and they have a stronger ability to handle reducing the organic matter that is produced by biological metabolism [47]. Then, in terms of biological factors, *M. meretrix* filter suspended particles in the water through their siphons to complete feeding, and large grain size substrates have larger gaps, making it easier for bacteria and microorganisms to invade, and increasing the likelihood of bacterial infestation during clam filter feeding [48]. Also, the small grain size substrate can help clams to remove attached organisms during sand diving, and the remaining attached organisms will suffocate and die due to the low level of oxygen in the substrate. Furthermore, bivalves with attached organisms will have difficulty in completing dives in large grain size substrates, making it difficult to remove the hazards caused by attached organisms [49].

### 4.3. Behavioral Characteristics

Burrowing in mudflat-dwelling mollusks is a heavily energy-intensive process. For example, *Mya arenaria* consumes an average of 7% of its energy reserves per incident of burrowing [50]. Therefore, mudflat-dwelling mollusks need to explore and collect information about their surroundings before diving into the sand to ensure that their first dive location meets their survival needs as the re-selection of habitat will inevitably result in unnecessary energy consumption [3]. The substrate selection process of juvenile *Percnon gibbesi* has three main behavioral manifestations, selection, exploration, and hesitation, and when in the vicinity of a preferred substrate, they have a short time to explore, hesitate, and make their choice [1]. This supports our experimental results. During the substrate selection process, *M. meretrix* took less time to select fine sand, and most individuals would burrow directly after coming into contact with fine sand, with less time spent hesitating (staying in place but extending their siphons and foot) and exploring (exploring a larger area through undirected movement), but individuals who selected coarse sand needed a longer time for hesitation and exploration. The phenomenon of hesitation and exploration during substrate selection can have positive effects, as longer hesitation will help them to obtain more information about the area to weigh the positives and negatives, and exploration will allow them to obtain information about a larger area and thus, they will have a higher chance of finding a suitable habitat.

Although zoobenthos are buried for a long time, they can move and accomplish most of their life activities through different forms of movement, such as migrating, foraging, and avoiding enemies [40,51]. For example, *Unio crassus* moves from the deepest parts of the river to the shore, where the small grain size of the substrate provides safer shelter [52]. Additionally, sea urchins leave their habitats in search of food when faced with food shortages [53]. *M. meretrix* also have a migration habit of moving from the intertidal zone to the subtidal zone in response to changes in the water environment or growth habits, and when they are faced with survival pressure due to unsuitable environments, they will find suitable habitats through their movement behavior [21,23]. Furthermore, in the natural environment, less fertile *M. meretrix* are more likely to exhibit mobile behavior, as a poor nutritional status means that the current habitat may lack food, or have a high predation pressure, unsuitable temperature, or salinity, and they need to migrate to find more suitable habitat [54]. Therefore, the movement behavior of *M. meretrix* can be regarded as an effective survival strategy for individuals under external environmental stress, and movement behavior can help them to avoid the harsh survival environment and to meet the survival needs of their different life stages [15,42]. In this study, *M. meretrix* showed a high percentage of movement in coarse sandy substrate, which may mean that coarse sandy substrate is not a suitable habitat for *M. meretrix*.

## 5. Conclusions

This study shows that juvenile *M. meretrix* has a higher preference for and stronger burrowing ability in fine sand (63–250 μm). As the substrate grain size increases, the burrowing ability and preference of *M. meretrix* decrease, and the bivalves show behavioral characteristics such as a prolonged selection time and an increased percentage of movement during substrate selection. In conclusion, sea areas or ponds with fine sand as the main grain size composition are more suitable for juvenile *M. meretrix* stock enhancement, and these results provide basic data for habitat selection and suitability evaluations for the aquaculture of *M. meretrix*. We recommend that it is more effective for stocking to find areas with a substrate particle size of 63–250 μm in the sea areas where *M. meretrix* resources have been greatly reduced due to overexploitation, and it can increase the success of *M. meretrix* resource recovery. The present study evaluated the essential habitat requirements of juvenile *M. meretrix* by observing its preference and behavior for different grain size substrates, and this could provide an additional approach for assessing the fundamental and realized niches for other zoobenthos.

## Figures and Tables

**Figure 1 animals-12-02094-f001:**
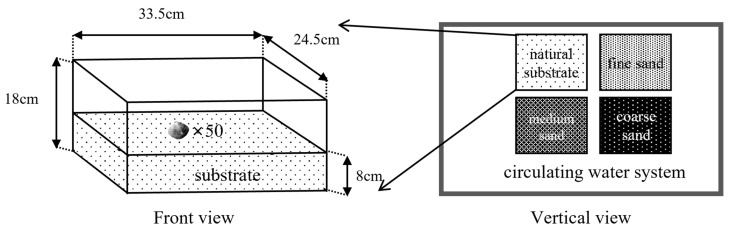
The experimental device for the burrowing ability.

**Figure 2 animals-12-02094-f002:**
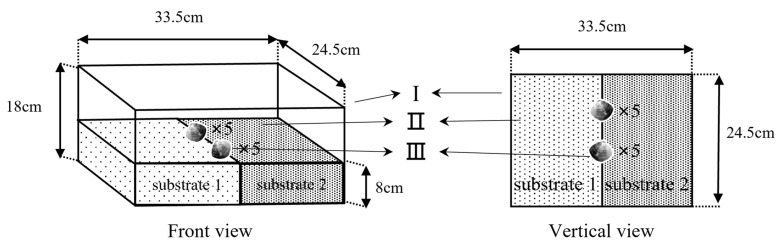
The device for substrate pairwise combinations. Unit Ι is the polypropylene plastic box. The different types of substrates (Ⅱ) are separated by a partition (Ⅲ).

**Figure 3 animals-12-02094-f003:**
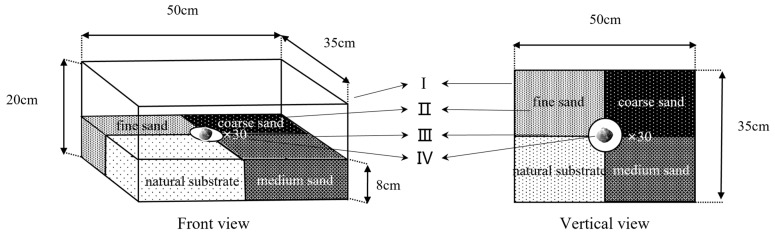
The device for the four substrate combinations. Unit Ι is the polyvinyl chloride sink, the four substrates (II) are separated by a partition (III), and a transparent plastic disc (Ⅳ), with a diameter of 8 cm and the same height as the substrate is fixed in the middle.

**Figure 4 animals-12-02094-f004:**
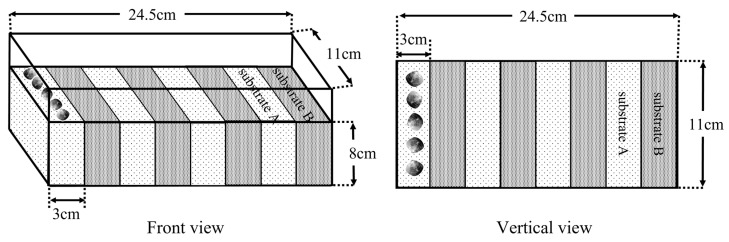
The device diagram of the experiment Ⅳ.

**Figure 5 animals-12-02094-f005:**
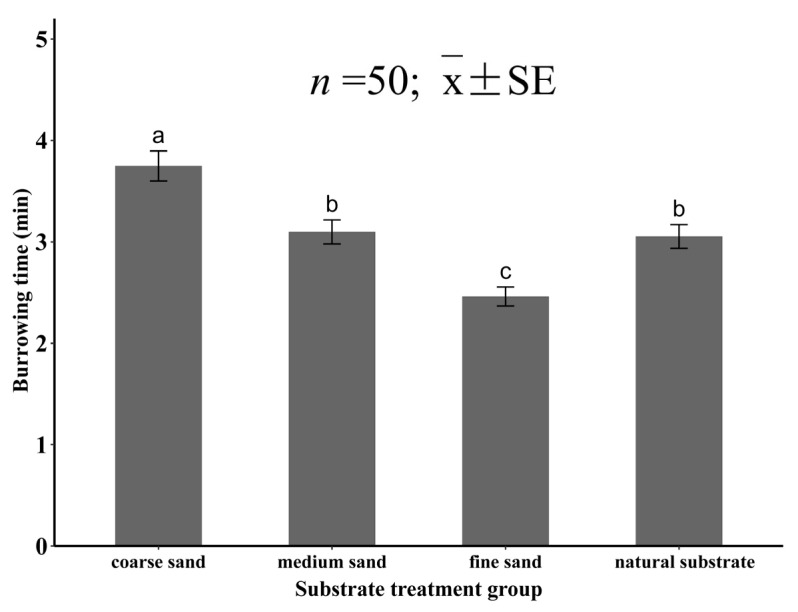
The burrowing time of juvenile *Meretrix meretrix* in different grain size substrates. The different letters above the bar graphs indicate significant differences (*p* < 0.05).

**Figure 6 animals-12-02094-f006:**
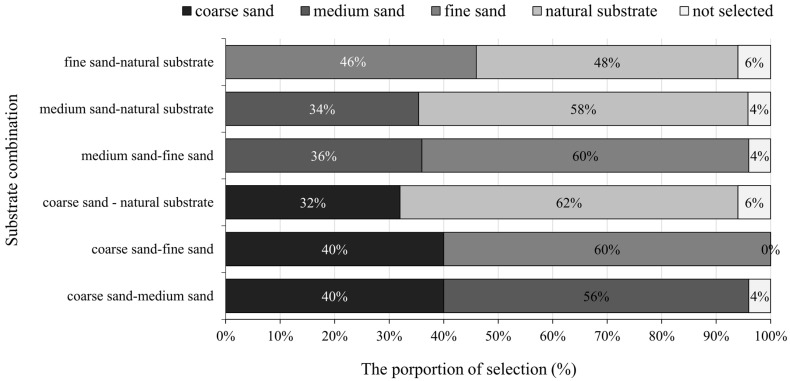
The preference of juvenile *Meretrix meretrix* for different grain size substrates in different combinations.

**Figure 7 animals-12-02094-f007:**
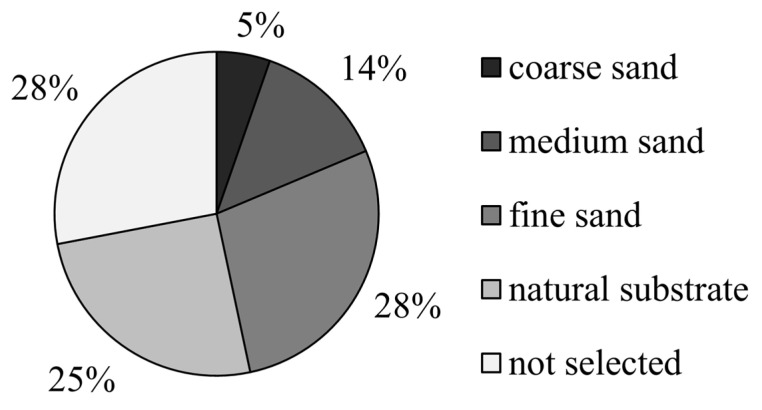
The preference of *Meretrix meretrix* for four grain size substrates.

**Figure 8 animals-12-02094-f008:**
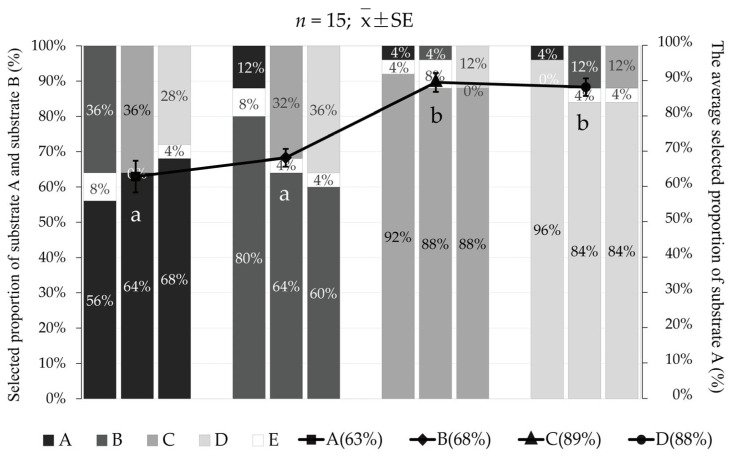
The preference of *Meretrix meretrix* for four grain size substrates: (A) coarse sand; (B) medium sand; (C) fine sand; (D) natural substrate; and (E) unselected. The bar graph shows the proportion selected of the different grain size substrates that were selected in twelve combinations. The line graph represents the average proportion of selection for the four grain size substrates. The different letters below the line graph indicate significant differences (*p* < 0.05).

**Figure 9 animals-12-02094-f009:**
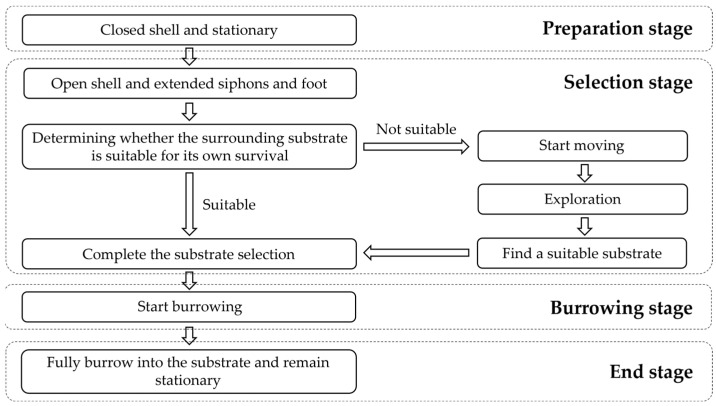
The different stages in the substrate selection process of *Meretrix meretrix*.

**Figure 10 animals-12-02094-f010:**
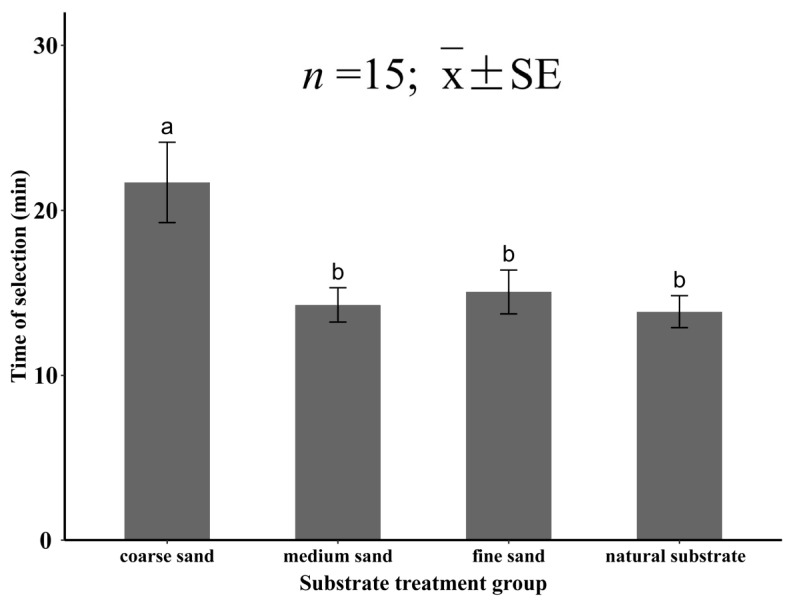
The selection time of juvenile *Meretrix meretrix* for different grain size substrates. The different letters above the bar graphs indicate significant differences (*p* < 0.05).

**Figure 11 animals-12-02094-f011:**
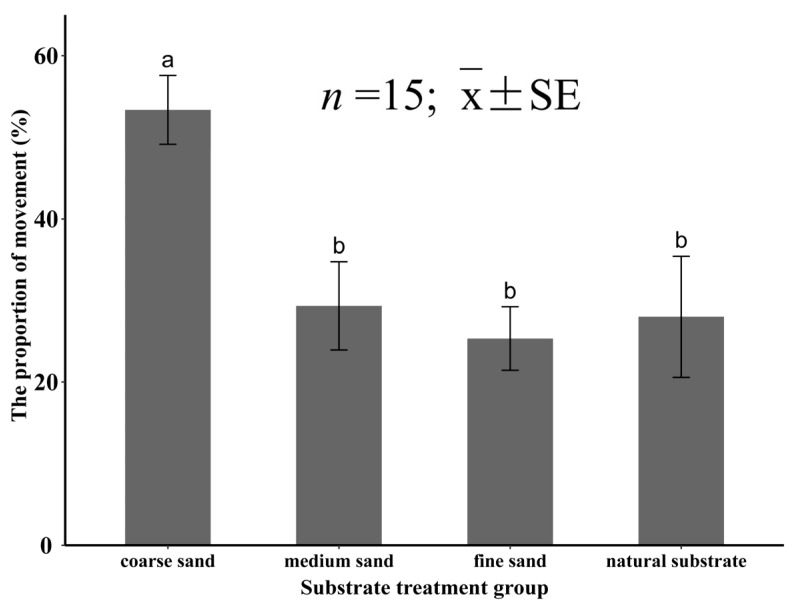
The percentage of juvenile *Meretrix meretrix* movement in the different grain size substrates. The different letters above the bar graphs indicate significant differences (*p* < 0.05).

**Table 1 animals-12-02094-t001:** Particle size classification of the natural substrate.

Grain Group Type	Fine Gravel	Coarse Sand	Medium Sand	Fine Sand	Silt	Mud
Particle diameter (μm)	>2000	2000–500	500–250	250–63	63–4	4–0
Volume proportion (%)	0.00	0.63	1.58	91.72	4.69	1.49
Median particle size	127 μm					

**Table 2 animals-12-02094-t002:** The twelve combinations for the substrates.

Substrate Name	Combination											
Substrate A	a	a	a	b	b	b	c	c	c	d	d	d
Substrate B	b	c	d	a	c	d	a	b	d	a	b	c

Note: a, coarse sand; b, medium sand; c, fine sand; and d, natural substrate.

## Data Availability

All data underlying this article are available in the main publication and in its Appendix A.

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
