# Peer review of "Influences of Substrate Grain Size on the Burrowing Behavior of Juvenile *Meretrix meretrix"

_animals, 2022, doi:10.3390/ani12162094_

Round 1
Reviewer 1 Report
This research is of particular interest given that M. meretrix has considerable commercial value, in fact this clam is one of the most consumed bivalve mollusc in China.
The results obtained in this study provide important information on the nature of the substrate suitable for settlement of M. meretrix reared juveniles.
Despite the existence in China of an active repopulation of M. meretrix through the settlement of juveniles reared in aquaculture, the stock shows signs of evident suffering.
The breeding of juveniles in good health together with the choice of the right substrate (grain size, habitat structure, etc.) are the basis for a correct restoring of the stock of M. meretrix.
The paper is well structured, the experimental design is appropriate. The methodologies as well as statistical tools used are suitable to achieve the objectives of the research.
The results are clear and appropriately interpreted.
The iconography is adequate. The figures properly show the data and are easy to interpret and understand.
Discussion and conclusion are thorough and well argued.
Author Response
This research is of particular interest given that M. meretrix has considerable commercial value, in fact this clam is one of the most consumed bivalve mollusc in China.
The results obtained in this study provide important information on the nature of the substrate suitable for settlement of M. meretrix reared juveniles.
Despite the existence in China of an active repopulation of M. meretrix through the settlement of juveniles reared in aquaculture, the stock shows signs of evident suffering.
The breeding of juveniles in good health together with the choice of the right substrate (grain size, habitat structure, etc.) are the basis for a correct restoring of the stock of M. meretrix.
The paper is well structured, the experimental design is appropriate. The methodologies as well as statistical tools used are suitable to achieve the objectives of the research.
The results are clear and appropriately interpreted.
The iconography is adequate. The figures properly show the data and are easy to interpret and understand.
Discussion and conclusion are thorough and well argued.
Author reply: Thank you very much for your positive comment. We will further develop our research on M. meretrix in behavioral and physiological metabolic.
Reviewer 2 Report
Dear Authors,
the manuscript is well presented. I made minor changes/comments in the annotated pdf file hoping to improve the quality of the study.
Sincerely yours.

Author Response
The manuscript is well presented. I made minor changes/comments in the annotated pdf file hoping
Author reply: Thank you very much for your valuable comments! We revised the manuscript according to your comments. The detail revisions are shown in the MS by “Track Changes” model.

Reviewer 3 Report
Dear Authors,
The manuscript is well prepared, with a structured experimental design and statistical analysis. I consider that the manuscript needs few adjustments and additions, in order to improve and make it viable for publication. Suggestions are in text boxes throughout the attached file and the inclusion of photos, as I have suggested, will enhance the quality of the manuscript. I have outlined the following comments that should be addressed:
-Description of fishing and its economic importance;
-Inclusion of map and photos in Material and Methods;
-Establish the standard used for the classification of substrates;
-And, the importance of rehabilitation actions and/or restoration of habitats for the species studied.

Author Response
The manuscript is well prepared, with a structured experimental design and statistical analysis. I consider that the manuscript needs few adjustments and additions, in order to improve and make it viable for publication. Suggestions are in text boxes throughout the attached file and the inclusion of photos, as I have suggested, will enhance the quality of the manuscript. I have outlined the following comments that should be addressed:
-Description of fishing and its economic importance;
-Inclusion of map and photos in Material and Methods;
-Establish the standard used for the classification of substrates;
-And, the importance of rehabilitation actions and/or restoration of habitats for the species studied.
Author reply: Thank you very much for your positive comment. We have revised your comments in detail as follows.
Moderate comment:
Introduction:
Point 1: Authors should provide a more detailed description of the study species, including a map showing its geographic distribution and photos.
Response 1: Thank you very much for the excellent suggestion. We have added a description of the morphological characteristics and geographical distribution of M. meretrix: “the shells have a triangular ovoid appearance and are mostly yellow and brown (Figure S1) [21]. M. meretrix is widely distributed along the coast of China, with abundant resources in the Liaoning, Shandong, Jiangsu, and Fujian Provinces.[22,23].” (see lines 119-122 in the revised manuscript). And we put a photo (Figure S1) of M. meretrix in the attachment files to enrich the manuscript.
We reviewed a large amount of literature, and unfortunately, in all of them, they describe the geographic distribution of the M. meretrix in textual form and do not show its geographic distribution in map form. Because M. meretrix are widely distributed along the coastal and estuarine areas of the western Indian Ocean (The Gulf of Oman and Aden) and Western Pacific (China, Korea, Vietnam and Japan), and it is found along the coast of China from north to south. Because its distribution is so wide, it is difficult to show its geographical distribution comprehensively and precisely in the form of a map, which may cause misunderstanding among readers, so we have described it in textual form in this paper, and we hope that we can get your approval.
References are as follows:
- Wang, R.; Wang, Z.; Zhang, J. Science Of Marine Shellfish Culture; Qingdao Ocean University Press: Qingdao, China, 1993; pp. 396-410.
- Zhang, A.; Li, H.; Yang, X.; Wang, L.; Gao, Y.; Song, M.; Yuan, X. Stock Assessment of Hatchery-Released Clam Meretrix meretrix in an Estuary of China From the Perspectives of Population Ecology and Genetic Diversity. Frontiers in Marine Science 2021, 8, 725238, https://doi.org/10.3389/fmars.2021.725238.
- Hashiguchi, M.; Yamaguchi, J.; Henmi, Y. Distribution and movement between habitats with growth of the hard clam Meretrix lusoria in the Shirakawa–Midorikawa estuary of the Ariake Sea. Fish. Sci. 2014, 80, 687-693, https://doi.org/10.1007/s12562-014-0748-4
Point 2: It is necessary for the authors to describe the main aspects of the economic importance of the species being studied, such as annual production, value (US$), estimate of people involved in this fishing activity, forms of exploitation and commercialization, etc.
Response 2: We have added a description of the economic importance of M. meretrix in the manuscript: ” It is one of the main commercial bivalves in China, and the annual production is approximately 3.5 × 105 – 4.0 × 105 t, accounting for more than 90% of the world production [24]. For example, the average annual production of M. meretrix in a town in Zhejiang Province is about 500 kg/667 m2, with a total annual production of 3,200 t and an export price of up to $2,500 per t, which has good economic benefits [25].” (see lines 122-127 in the revised manuscript).
References are as follows:
- Chen, L.; Wang, X.; Chen, D. Research on the development of hard clam market in China mainland. J. Fish. Univ. Shanghai 2004, 13, 283-287.
- Ji, B. Analysis of the cost benefit of exporting marine aquaculture products from Wenzhou with Meretrix meretrix as an example. J. Aquac. 2014, 35, 37-40, https://doi.org/10.3969/j.issn.1004-2091.2014.03.016.
Point 3: It is necessary to make a brief description of the cultivation system of the species, in order to understand where the importance of this study is located (at what stage of production).
Response 3: We have added a description of the cultivation system of M. meretrix: ” The artificial culture process of M. meretrix is divided into parental maturation promotion and spawning, larvae cultivation, intermediate cultivation of juvenile M. meretrix, mudflat or pond stocking, and culture [32]. Thus, clarifying the substrate adaptability of M. meretrix can provide a reference for habitat selection during intermediate pond cultivation and stock enhancement in the mudflats.” (see lines 186-189 in the revised manuscript).
References are as follows:
- Zhang, A.; Li, T.; Su, X.; Liu, B. Current status and prospect of Meretrix meretrix culture. Fish. Sci. 2005, 24, 31-33, https://doi.org/10.16378/j.cnki.1003-1111.2005.02.011.
- Materials and Methods
2.1 Acquisition of experimental animals and substrates
Point 4: Here, cite the Statement of the Ethics Committee, which can be found in the supplementary files.
Response 4: We cited the Statement of the Ethics Committee in the manuscript: “The ethical regulations concerning the use of experimental animals were followed (see the Statement of the Ethics Committee in the supplementary files).” (see lines 193-194 in the revised manuscript).
Point 5: Which Standard (see for example: https://www.civilengineeringforum.me/soil-particle-size/) was used to establish the size groups? Authors should report which standard was adopted. A photo showing each group will enrich the manuscript.
Response 5: We have added the particle size classification standard that we use: ” Then, it was dried and sieved into coarse sand (grain size of 500–2000 μm), medium sand (grain size of 250–500 μm), and fine sand (grain size of 63–250 μm) based on the Wentworth scale [33].” (see lines 208-209 in the revised manuscript).
We put a photo (Figure S2) of four substrates in the attachment files to enrich the manuscript, and cited in the manuscript: ” Four substrates with different grain sizes were used in this study (Figure S2 provides a picture of the four substrates),” (see lines 210 in the revised manuscript).
References are as follows:
- Doeglas, D.J. Grain-Size Indices, Classification and Environment. Sedimentology 1968, 10, 83-100, https://doi.org/10.1111/j.1365-3091.1968.tb01101.x.
2.3 Experiment Ι
Point 6: Although Figures 1, 2 and 3 show the experimental design clearly, I recommend that the authors first insert a picture of the entire experimental apparatus (with the box, video camera, etc.) in the laboratory. This will enrich the manuscript.
Response 6: We put a photo (Figure S3) of the entire experimental apparatus in the attachment files to enrich the manuscript, and cited in the manuscript: ” (Figure S3 provides a picture of the entire experimental apparatus in the laboratory)” (see lines 219-220 in the revised manuscript).
5 Conclusions
Point 7: In natural environments, where the species is exploited, would it be possible for the study to recommend the REHABILITATION or RESTORATION of these areas, considering the results (sediments to be used) obtained in the study? The authors could make these recommendations in the Conclusion.
Response 7: Thank you very much for the advice, it was very helpful for improving our article. Following the reviewer’s suggestion, we have added a sentence: ” We recommend that it is more effective for stocking to find areas with a substrate particle size of 63–250 μm in the sea areas where M. meretrix resources have been greatly reduced due to overexploitation, and it can increase the success of M. meretrix resource recovery.” (see lines 639-642 in the revised manuscript).
